# Green Extraction of Six Phenolic Compounds from Rattan (*Calamoideae faberii*) with Deep Eutectic Solvent by Homogenate-Assisted Vacuum-Cavitation Method

**DOI:** 10.3390/molecules24010113

**Published:** 2018-12-29

**Authors:** Qin Cao, Junhan Li, Yu Xia, Wei Li, Sha Luo, Chunhui Ma, Shouxin Liu

**Affiliations:** Key Laboratory of Bio-Based Material Science and Technology, Ministry of Education, College of Material Science and Engineering, Northeast Forestry University, Harbin 150040, China; nefucaoqin@163.com (Q.C.); nefulijunhan@163.com (J.L.); xiayu0712@163.com (Y.X.); liwei19820927@126.com (W.L.); luo.sha.85@163.com (S.L.)

**Keywords:** *Rattan* (*Palmae*), phenolic compounds, deep eutectic solvent (DES), homogenate-assisted vacuum-cavitation extraction (HVE), RP-HPLC

## Abstract

A homogenate-assisted vacuum-cavitation extraction (HVE) method with a “green” solvent (a deep eutectic solvent, DES) was developed to extract phenolic compounds from rattan (*Calamoideae faberii*). In this study, the optimum molar ratio of choline chloride (ChCl) and ethylene glycol (EG) was 1:3, the optimum volume ratio of ChCl-EG:H_2_O was 6:4, the solid-liquid ratio of HVE was 1:15, and the extraction time of homogenate and vacuum-cavitation were 2.0 min and 25 min, respectively. Under the optimum parameters of HVE, the extraction yield of total phenolic content with ChCl-EG solution was 6.82 mg/g. The higher total phenolic content was detected in fruit tissues (seeds 81.24 ± 1.55 mg/g, episperm 43.21 ± 0.87 mg/g, and arillus 38.47 ± 0.74 mg/g), followed by in leaves (sheath 19.5 ± 0.38 mg/g and blade 17.81 ± 0.33 mg/g). In addition, the content of specific phenolic compounds in aqueous and DES extracts was determined. Chlorogenic acid was the most abundant phenol in most organs of the rattan plant. Gallic acid was mainly distributed in the arillus; protocatechuic acid was mainly distributed in the arillus, sheath, and blade; protocatechuic aldehyde was mainly distributed in the blade, seed, and sheath; (+)-catechins were mainly distributed in the episperm, seed, and sheath; and epigallocatechin gallate was mainly distributed in the blade. The recovery rates of gallic acid, protocatechuic acid, protocatechuic aldehyde, (+)-catechins, chlorogenic acid, and epigallocatechin gallate were 93.77%, 94.09%, 97.32%, 97.83%, 94.41%, and 92.47%, respectively, by AB-8 resin.

## 1. Introduction

As a class of natural “green” solvents, deep eutectic solvents (DESs) have attracted increased research attention during recent years because of their excellent characteristics, and represent a cheap alternative to ionic liquids (ILs) [1,2,3]. DESs can be used in a wide range of applications such as organic synthesis, catalysis, electrochemistry, and nanotechnology [4,5,6]. Moreover, DESs are suitable for extraction processes because of their negligible volatility at room temperature, nontoxicity, non-reactivity with water, biodegradability, and low environmental and economic impact [7,8,9]. Meanwhile, the rate of articles being published on natural-products separation is increasing. In the past five years, a few dozen interesting papers focusing on cleaner production procedures based on both solid–liquid extraction using DESs and homogenate-assisted vacuum-cavitation extraction have appeared.

In recent years, homogenate technology has been used to extract active compounds from various materials [10]. In the homogenization process, plant materials are pulverized using strong mechanical force, and the liquid shear force in the material–solvent mixture allows the solvent to penetrate into the material easily, thereby facilitating the fast dissolution of target compounds. In a homogenate treatment, material pulverization and solvent extraction are completed in one step. Therefore, this method has many advantages in a cleaner production process, including short operation time, no powder dust pollution, low extraction temperature, and high efficiency.

Recently, researchers in the fields of analytical chemistry and sample preparation have focused their attention on the cavitation effect, which can be classed into acoustic cavitation [11] or hydrodynamic cavitation [12] according to the mode of cavity formation [13].

The benefits of ultrasonic cavitation in extraction are the intensification of mass transfer and cell disruption, improved penetration, and capillary effects [14,15]. However, the use of high-frequency ultrasonic waves for cavitation leads to an increase in the solvent temperature. This increases energy consumption and limits the scale of industrial production. Recently, therefore, negative-pressure cavitation extraction and hydrodynamic cavitation, which require less energy and have lower energy dissipation, have been used in sequence to extract natural products from plants in a closed cleaner production process. The vacuum-cavitation extraction method has been used to extract polysaccharides [16], alkaloids [17], isoflavones [18], stilbenes [19], and seed oil [20] from plant materials.

Rattan is a barbed climbing monocotyledonous plant in the family Palmae (Angiospermae, Monocotyledoneae, Principes) [21]. It is one of the most important representatives of this branch in the tropics, and is popularly known as “green gold”. Wild rattan is distributed in Indonesia, the Philippines, Malaysia, and Thailand, and is widely cultivated in the tropical Americas, tropical Asia, and in the Pacific Islands [22]. The plant taxonomist Dransfield separated the Palmae into 13 genera, with about 610 species and subspecies known around the world [22]. Previous studies have mainly focused on the commercial aspects of rattan cane as a raw material for furniture and handicraft industries [23], rather than its chemical composition or biological activities. In general, only the cane part of rattan is used, and the other parts are waste materials. The aim of this study, therefore, was to extract phenolic compounds from rattan waste parts (blade, sheath, episperm, arillus, seed, and stem). There are two benefits of this approach: it uses rattan waste resources, and provides phenolic compounds for use as active ingredients.

Phenolic compounds can be divided into several classes, including phenolic acids and their derivatives, dehydrodiferulates and dehydrotriferulates, flavan-3-ol monomers, and others [24]. At present, the main research reports of phenolic compounds are gallic acid, protocatechuic acid, protocatechuic aldehyde, catechin, chlorogenic acid and epigallocatechin gallate. Their antioxidant [25,26,27] and free-radical scavenging activities [28,29,30] re excellent for a wide range of industrial applications, such as food additives and colorants, which can be attributed to the phenolic hydroxyl groups they possess. 

In this study, a DES extraction method combining homogenate technology with vacuum-cavitation digestion was used to accelerate the extraction of phenolic compounds from rattan waste materials (*Palmae*, *Calamoideae*, *Trachycarpus*). The extraction mechanism of homogenate-assisted vacuum-cavitation with deep eutectic solvent was studied in detail, and the innovation lies in the formation of hydrogen bonds in DES that decrease the barrier of plant cell walls recalcitrance. And this effect was worked synergistically with homogenization extraction and negative-pressure cavitation extraction. In addition, a reversed-phase high-performance liquid chromatography (RP-HPLC) method was used to quantify six phenolic compounds simultaneously. Finally, the content distribution of the six different phenolic compounds in different parts of the rattan plant was investigated.

## 2. Results and Discussion

### 2.1. Chemical Analysis of Rattan Materials 

The chemical composition of rattan (100.0 g) was as follows: cellulose (9.56% ± 0.26%), hemicellulose (50.97% ± 0.65%), lignin (6.57% ± 0.07%), aqueous extracts (which was extracted with pure water) (3.5% ± 0.08%), and ethanolic extracts (9.85% ± 0.2%). The content of phenolic compounds in the aqueous extract (extracted by heating and stirring with water for 6 h) was 0.74% ± 0.01%. The content of phenolic compounds in the ethanolic extracts (extracted by Soxhlet extraction for 6 h with 80% ethanol solution) was 2.29% ± 0.01%. 

### 2.2. RP-HPLC Analysis of Phenolic Compounds

For the HPLC-UV analysis, methanol-water-phosphoric acid (10:88:2, *v*/*v*/*v*) was used as the mobile phase at a flow rate of 1.0 mL/min. The injection volume was 10 µL and the column temperature was 25 °C. Precision the standard 10.0 mg, and then set capacity to 10 mL, obtained the single standard reserve solution. And then mixed the single standard reserve solution, obtained the mixing standards reserve solution (0.167 mg/mL for each phenolic standard). Finally, diluted the mixing standards by 2 times in turn, and were detected by measuring absorbance at 274 nm during a run time of 60 min (Figure 1). Table 1 shows the retention time, the corresponding calibration curves, the limit of quantity (LOQ) and the limit of detection (LOD) for gallic acid, protocatechuic acid, protocatechuic aldehyde, (+)-catechins, chlorogenic acid, and EGCG.

### 2.3. Screening of DES Solution 

#### 2.3.1. Effect of DES Composition on Extraction Efficiency of Phenolic Compounds

DESs are composed of a hydrogen bond acceptor (HBA) and a hydrogen bond donor (HBD), in contrast to ILs, which consist of anions and cations. The interactions between HBA and HBD involve mostly hydrogen bonding, occasional electrostatic forces, and van der Waals interactions. The most important intramolecular bonds in DESs are the hydrogen bonds between the HBD and halide anions [3]. Through the preliminary experiments, Choline chloride (ChCl) as a kind of hydrogen bond acceptor, polyol as hydrogen bond donor showed a better extraction efficiency of phenolic acid compounds. Thus, in this study, the polyol (ethylene glycol (EG), glycerol (GI) and 1,4-butanediol (BDO)) as hydrogen bond donor was screened. And the extraction yield of the total phenol content as the response value, the different DESs (ChCl-EG, ChCl-GI, ChCl-BDO) with the same volume ratio of DES:H_2_O (6:4) was investigated. The results were shown in Figure 2. In Figure 2a, the extraction yields when using ChCl-EG, ChCl-Gl, and ChCl-BDO with different molar ratios (1:1, 1:2, 1:3, 1:4, 1:5) are displayed. In all three DESs, hydrogen bonds are shared between the HBD (polyol) and chloride ions from ChCl (HBA), as showed in Appendix A, for polyols, there are more than two hydroxyl groups in the molecule that can form large molecules by forming intramolecular hydrogen bonds, and can also form intermolecular hydrogen bonds with ChCl. The greater number of OH was provided by GI, when the same molar ratios of polyol, but the better water absorbent performance of GI may form the intermolecular hydrogen bonds, resulting in a decrease in the number of hydrogen bonds with ChCl. Compared with BDO, the steric effects to the formation of intermolecular hydrogen bonds with is ChCl smaller. Therefore, the extraction yield of total phenolic compounds with ChCl-EG (6.97 ± 0.11 mg/g) was higher than that with ChCl-Gl (4.79 ± 0.13 mg/g) and ChCl-BDO (4.77 ± 0.14 mg/g), and 1:3 was the optimum molar ratio of ChCl and polyol. 

#### 2.3.2. Effect of Water Content in DES on Extraction Efficiency of Phenolic Compounds

The main disadvantage of DESs is their high viscosity, which not only hinders the mass transport from plant matrices to solution but also leads to handling difficulties (e.g., in filtration, decantation, and dissolution). Thus, the addition of water to DES was a vital factor affecting the extraction capacity of the target compounds. From Figure 2b, with the volume ratio of ChCl-EG:H_2_O decreasing, the extraction efficiency of total phenolic compounds firstly increased, and declined gradually thereafter (Appendix A). The highest extraction yield (6.97 ± 0.11 mg/g) was achieved with 40% water (volume percentage) in ChCl-EG solution. The addition of water decreases the viscosity, which increases the osmotic effect of DES solution, enhancing the mass transfer from plant matrices to solution, and thus resulting in boosted extraction efficiency. However, excessive water addition can increase the chemical polarity of DES solution, and the dissolve of phenolic compounds increased according to the “similar solubility principle”. Therefore, the volume ratio of ChCl-EG:H_2_O = 6:4 was used for further extraction processes.

### 2.4. Factors Affecting Total Phenolic Content after Homogenate-Assisted Extraction

#### 2.4.1. Solid-Liquid Ratio

During the extraction step, an excessively high solvent content may complicate the extraction process, create unnecessary wastage, and increase energy consumption during recycling, while an excessively low solvent content may lead to incomplete extraction. To evaluate the effect of the solid-liquid ratio during extraction, 20.0 g dried rattan material was mixed with DES and aqueous solution at a range of different solid-liquid ratios (1:5, 1:8, 1:10, 1:12, 1:15, and 1:20 g/mL), and then the samples were homogenized for 2.0 min. As shown in Figure 3, the total phenolic content increased with increasing solvent volume up to a solid-liquid ratio of 1:15 (total phenolic content, 8.41 ± 0.14 mg/g with aqueous solution; 8.87 ± 0.16 mg/g with DES), but did not increase significantly as the solid-liquid ratio increased further. Therefore, considering the need to extract the total phenolic content while minimizing the energy consumption of solvent recovery, the 1:15 solid-liquid ratio was used for further HAE experiments.

#### 2.4.2. Duration of Homogenate-Assisted Extraction

To investigate the effect of extraction time on the total phenolic content after homogenate extraction, 20.0 g dried sample was mixed with DES solution or pure water (ratio, 1:15), and then the homogenate was extracted for 0.5, 1.0, 1.5, 2.0, 2.5, or 3.0 min. As shown in Figure 4, as the extraction time increased, the total phenolic content initially increased. After homogenate extraction for 2.0 min (total phenolic content, 1.97 ± 0.04 mg/g with pure water; 8.75 ± 0.16 mg/g with DES), there were no further increases in the total phenolic content. Therefore, 2.0 min was selected as the optimal duration time for homogenate extraction of phenolic compounds.

#### 2.4.3. Homogenate-Assisted Extraction Mechanism

The homogenate-assisted extraction process includes the pulverization of raw material, and the mixing of the solid (plant material) and the liquid (extraction solvent) phases. In essence, it is a solid–liquid mass transfer process enhanced using an external force [10]. In Figure 5, the formation of DES liquid film on the plant cell surface was analyzed, from a microscopic point of view. Three effects are involved in homogenate-assisted extraction process: (1) The shearing action reduces the size of the large solid particles and destroys most of the plant cell walls. At the same time, a liquid membrane layer which contained more hydrogen bonds forms at the cell surface, the permeability of cell was increased and the phenolic acids are released into the solvent; (2) Shearing of solid particles into smaller pieces, leading to an increase in the specific surface area of the solid–liquid phase interface, and an increase in the area of the liquid membrane attached to the solid surface. This increases the mass transfer rate from solid to liquid, and shortens the extraction time; and (3) Partial disappearance of the liquid membrane. During the homogenization process, the thickness of the liquid membrane (also known as the dynamic membrane) changes as the stirring speed changes, becoming thinner at higher stirring speeds. In addition, the dissolution rate of target components in plant cells increases with stronger turbulence in the mixture. Therefore, the homogenate extraction process is actually an unstable and non-equilibrium process by which active components are released from plant cells and dissolved into the solvent [31,32]. In short, as the homogenate-assisted extraction is prolonged, the plant particle size becomes smaller, and the smaller particle size can provide a higher specific surface area, thereby shortening the mass transfer path and improving mass transfer efficiency.

### 2.5. Factors Affecting Cavitation Extraction

#### 2.5.1. Vacuum-Cavitation Time

The generation of vacuum-cavitation by negative pressure is a cheap and energy-efficient method. As shown in Figure 6, extractions were carried out at negative pressure with a 180 W vacuum pump for 5, 10, 15, 20, 25, 30, 35, 40, and 45 min. The total phenolic content increased significantly as the treatment time increased from 0 to 25 min (total phenolic content after 25 min vacuum-cavitation, 21.44 ± 0.44 mg/g with pure water; 90.33 ± 1.89 mg/g with DES). However, the content increased only slightly as the vacuum-cavitation treatment time was further increased from 25 min to 45 min (total phenolic content after 45 min vacuum-cavitation, 23.65 ± 0.38 mg/g with pure water; 96.47 ± 2.02 mg/g with DES). Therefore, 25 min was used as the vacuum-cavitation time in further extraction experiments.

#### 2.5.2. Negative-Pressure Cavitation Effect

Next, after optimizing the extraction times and volumes as described above, a comparative study was conducted to compare the vacuum-cavitation effect with the well-known ultrasonic cavitation effect. Cavitation is a fluid mechanics phenomenon in which millions of tiny vapor bubbles sequentially form, grow, and collapse in the liquid phase or at the liquid–solid interface. As shown in Figure 7, with negative pressure, the formation and expansion of bubbles in the turbulent liquid, and the turbulence generated in the liquid–solid mixture, accelerated the collapse of bubbles, which was controlled by the vacuum power. The bubbles produced during ultrasonic extraction formed on the solid surface, while those produced under negative pressure grew faster and had a shorter collapse cycle. Moreover, there was a vigorous stirring effect of solvent which contained more hydrogen bonds under negative pressure, so that the surface of the cell walls broke down and the solvent readily diffused into the material, and hydrogen bonds that decrease the barrier of plant cell walls recalcitrance. In this system, rapid mass transfer was accomplished [33], and consequently, the target compounds were transferred from the matrix into the solvent [34]. As the treatment time increased from 0 to 20 min, the total phenolic content increased significantly. The cavitation extraction with negative pressure at room temperature can avoid being heated resulting in isomerization. After treatment 25 min, the total phenolic yield increased minimally.

### 2.6. Content Distribution of Total Phenolic in the Rattan Plant

The total phenolic contents in different parts of the rattan plant were determined using the Folin–Ciocalteu method. With DES and pure water as the extraction solvents, the optimized HVE method was used to extract phenolic compounds from rattan. The total phenolic contents in DES and aqueous extracts from various organs are shown in Figure 8. The highest total phenolic content was in fruit tissues, including the seeds (81.24 ± 1.55 mg/g in DES extracts; 8.75 ± 0.17 mg/g in aqueous extracts), episperm (43.21 ± 0.87 mg/g in DES extracts; 4.30 ± 0.08 mg/g in aqueous extracts), and arillus (38.47 ± 0.74 mg/g in DES extracts; 3.81 ± 0.07 mg/g in aqueous extracts); followed by the leaf tissues, including the sheath (19.5 ± 0.38 mg/g in DES extracts; 1.94 ± 0.04 mg/g in aqueous extracts) and blade (17.81 ± 0.33 mg/g in DES extracts; 1.79 ± 0.03 mg/g in aqueous extracts). The lowest total phenolic content was in the stem (5.11 ± 0.10 mg/g in DES extracts; 0.52 ± 0.01 mg/g in aqueous extracts).

### 2.7. Differences in Specific Phenolic Compound among Different Parts of Rattan

Six different phenolic compounds were simultaneously quantified in different parts of rattan using the RP-HPLC method, namely gallic acid, protocatechuic acid, protocatechuic aldehyde, (+)-catechins, chlorogenic acid, and EGCG. Chlorogenic acid was the most abundant phenolic compound in all organs except for the arillus (Figure 9). In the aqueous extract from the rattan blade, the concentrations of gallic acid, protocatechuic acid, protocatechuic aldehyde, (+)-catechins, chlorogenic acid, and EGCG were 1.356 ± 0.024 mg/g, 0.032 ± 0.001 mg/g, 0.055 ± 0.001 mg/g, 0.225 ± 0.004 mg/g, 18.919 ± 0.378 mg/g, and 1.768 ± 0.034 mg/g, respectively (Figure 9a). The highest EGCG content was in the blade (1.768 ± 0.034 mg/g in DES extracts; 1.440 ± 0.028 mg/g in aqueous extracts), while EGCG was present at only trace amounts in other organs (undetectable by HPLC). In the aqueous extracts from the rattan sheath, the concentrations of gallic acid, protocatechuic acid, protocatechuic aldehyde, (+)-catechins, and chlorogenic acid were 1.689 ± 0.031 mg/g, 0.097 ± 0.002 mg/g, 0.018 ± 0.001 mg/g, 1.132 ± 0.028 mg/g, and 5.785 ± 0.116 mg/g, respectively (Figure 9a). In the aqueous extracts of rattan episperm, only (+)-catechins and chlorogenic acid were detected, at concentrations of 8.52 ± 0.155 mg/g and 9.291 ± 0.172 mg/g, respectively. In the aqueous extract from the arillus, only gallic acid, protocatechuic acid, and chlorogenic acid were detected, at concentrations of 2.039 ± 0.040 mg/g, 0.071 ± 0.003 mg/g, and 0.320 ± 0.006 mg/g, respectively. In the aqueous extract from rattan seeds, the concentrations of gallic acid, protocatechuic aldehyde, (+)-catechins, and chlorogenic acid were 0.038 ± 0.003 mg/g, 0.046 ± 0.003 mg/g, 3.177 ± 0.061 mg/g, and 14.399 ± 0.288 mg/g, respectively. In the aqueous extracts from rattan stems, only gallic acid and chlorogenic acid were detected, at concentrations of 0.674 ± 0.018 mg/g and 5.209 ± 0.115 mg/g, respectively.

As shown in Figure 9b, in DES extracts from rattan blades, the concentrations of gallic acid, protocatechuic acid, protocatechuic aldehyde, (+)-catechins, chlorogenic acid, and EGCG were 0.163 ± 0.005 mg/g, 0.014 ± 0.001 mg/g, 0.099 ± 0.002 mg/g, 0.057 ± 0.002 mg/g, 34.414 ± 0.844 mg/g, and 1.440 ± 0.030 mg/g, respectively. The concentrations of gallic acid, protocatechuic acid, protocatechuic aldehyde, (+)-catechins, and chlorogenic acid in rattan sheaths were 0.241 ± 0.005 mg/g, 0.032 ± 0.002 mg/g, 0.085 ± 0.002 mg/g, 3.161 ± 0.084 mg/g, and 12.149 ± 0.296 mg/g, respectively. In DES extracts of rattan episperm, only (+)-catechins and chlorogenic acid were detected, at concentrations of 11.981 ± 0.299 mg/g and 13.538 ± 0.288 mg/g, respectively. Only gallic acid, protocatechuic acid, and chlorogenic acid were detected in DES extracts of the arillus, at concentrations of 2.465 ± 0.034 mg/g, 0.048 ± 0.001 mg/g, and 0.66 ± 0.020 mg/g, respectively. The concentrations of gallic acid, protocatechuic aldehyde, (+)-catechins, and chlorogenic acid in DES extracts from rattan seeds were 0.020 ± 0.001 mg/g, 0.093 ± 0.002 mg/g, 4.697 ± 0.051 mg/g, and 7.959 ± 0.081 mg/g, respectively. In DES extracts from rattan stems, only gallic acid and chlorogenic acid were detected, at concentrations of 0.029 ± 0.001 mg/g and 4.831 ± 0.050 mg/g, respectively.

### 2.8. Recovery of Phenolic Compounds from ChCl-EG

The use of the AB-8 macroporous resin to recover the phenolic compounds from ChCl-EG extracts was evaluated. The total recovery rate of all six phenolic compounds by dynamic chromatography at the upper flow rate of 0.5 mL/min was 94.4%. Then, ChCl-EG was removed with deionized water and decompression evaporation for reuse. The adsorbed phenolic compounds were then eluted with 70% ethanol solution at 1.0 mL/min, and the ethanolic fractions (each 10 mL) were collected for HPLC analysis. Finally, the ethanolic eluates containing each respective phenolic compound were combined, each combined eluate was dried with a vacuum evaporator, and the purity and recovery rate were calculated (Table 2). 

## 3. Materials and Methods

### 3.1. Materials 

#### 3.1.1. Rattan Materials

Rattan (Calamoideae faberii) materials (all the parts) were collected from Hainan, China, in October 2016 and were dried in a shaded, well-ventilated area, and then sheared into 1 cm × 1 cm square pieces. 

#### 3.1.2. Chemicals

Standards (>98% purity) of gallic acid, protocatechuic acid, protocatechuic aldehyde, (+)-catechins, chlorogenic acid, and epigallocatechin gallate (EGCG) were purchased from the National Institute for the Control of Pharmaceutical and Biological Products (Beijing, China). Folin–Ciocalteu reagent was obtained from Sigma (St. Louis, MO, USA). Other reagents included choline chloride (ChCl: 98%, Macklin Biochemical Co., Ltd., Shanghai, China), ethylene glycol (EG: 99%, Fuyu Fine Chemical Co., Ltd., Tianjin, China), glycol (Gl: 99%, Tiantian Chemical Testing Factory, Tianjin, China), 1,4-butylene glycol (BDO, 98%, Guangfu Fine Chemical Co., Ltd., Tianjin, China). Deionized water was purified using a Milli-Q Water Purification system (Millipore, Billerica, MA, USA). Methanol and phosphoric acid (HPLC grade) were purchased from J&K Chemicals Ltd. (Shanghai, China). All other solvents and chemicals were of analytical grade and were purchased from Beijing Chemical Reagents Co. (Beijing, China). All solutions used for HPLC were filtered through 0.22 μm membranes before use.

### 3.2. Methods

#### 3.2.1. Determination of Total Phenolic Content

The total phenolic content of the extracts was determined as described elsewhere [35], with modifications. The rattan extract was diluted with distilled water to an appropriate concentration, and then 1 mL of the diluted sample was mixed with 100 mL Folin–Ciocalteu reagent (previously diluted with water 1:1, *v*/*v*). A saturated sodium carbonate solution (10%, 2 mL) was then added, and the mixture was kept at room temperature for 60 min before being centrifuged at 3000 r/min for 5 min. The absorbance of the supernatant was measured at 725 nm.

#### 3.2.2. RP-HPLC analysis of Phenolic Compounds

The RP-HPLC system consisted of a Waters 717 automatic sample handling system, an HPLC system equipped with a 1525 binary pump, a 717 automatic column temperature control box, and a 2487 UV-detector (all from Waters, Milford, MA, USA). Chromatographic separation was performed on a HiQ sil-C18 reversed-phase column (4.6 mm × 250 mm, 5 μm, KYA TECH Corp., Tokyo, Japan) for the simultaneous determination of six phenolic compounds. 

#### 3.2.3. Preparation of DESs

DESs were synthesized by a heating method [36]. The hydrogen bond acceptor in each DES was ChCl. This was mixed with a hydrogen bond donor (one of the polyols listed in Section 2.3.1) in an appropriate molar ratio and placed in a reaction flask with magnetic agitation at 80 °C for 30 min, resulting in the formation of a transparent, homogeneous liquid. Three DESs with different compositions, including ChCl-EG (ethylene glycol), ChCl-Gl (glycerol), and ChCl-BDO (1,4-butylene glycol), were prepared using the three different hydrogen bond donors.

#### 3.2.4. Extraction of Phenolic Acid Compounds with DESs

A homogenate-assisted extraction (HAE) method was used to extract phenolic compounds from the rattan materials using the DESs. A 200 W homogenizer (HANUO-JJ2, Shanghai Hannuo Instrument Co. Ltd., Shanghai, China) was used to grind the materials, and 20.0 g of the ground material was added to 300 mL DES solution in the homogenizer. After homogenization for 2.0 min, the extract was cooled to room temperature and then filtered through a 0.45 µm filter before total phenolic content analysis.

#### 3.2.5. HVE Method to Extract Phenolic Acid Compounds

A homogenate-assisted vacuum-cavitation extraction (HVE) method was used to extract phenolic acid compounds from the rattan materials. A negative-pressure cavitation glass column (50 × 4.0 cm) was made in our laboratory and was connected to a 180 W circulating-water vacuum pump (SHB-III, Great Wall Scientific Industry and Trade Co. Ltd., Zhengzhou, China), to control the air flow rate, and the maximum vacuum can reach to −0.1 Mpa. The device scheme of experiments was according to our previous studies [31].

In the extraction experiments, 20.0 g material was added to 300 mL DES solution in the homogenizer. After homogenization for 2.0 min, the material and solvent were transferred to the negative-pressure cavitation glass column. After vacuum-cavitation extraction at room temperature, the extracts were filtrated through a 0.45 µm filter before total phenolic content analysis and HPLC analysis.

#### 3.2.6. Recycling of DESs

Absorption of the phenolic compounds from the DES extracts was conducted based on AB-8 macroporous resin (HaoJu resin Technology Co. Ltd., Tianjin, China) column chromatography [37]. The polar DES was removed with deionized water and decompression evaporation for reuse. The sample was then eluted with 70% ethanol solution, and the ethanolic fractions were dried with a vacuum evaporator, and redissolved with 10% methanol for HPLC analysis.

#### 3.2.7. Statistical Analyses

Results are expressed as mean values ± SD (*n* = 3). Mean values from different experiments were compared by ANOVA using Microsoft Excel software (Microsoft Corporation, Redmond, WA, USA). Differences at *p* < 0.05 were considered to be significant.

## 4. Conclusions

Homogenate-assisted vacuum-cavitation extraction (HVE) was used to extract phenolic compounds with deep eutectic solvents (DESs) from different parts of rattan (*Palmae*), and the extraction mechanisms of homogenate-assisted negative-pressure cavitation have been analyzed in detail. The optimum molar ratio of choline chloride (ChCl) and ethylene glycol (EG) was 1:3, the optimum volume ratio of ChCl-EG:H_2_O was 6:4, the solid-liquid ratio was 1:15, and the extraction time of homogenate and vacuum-cavitation were 2.0 min and 25 min, respectively. Under the optimum parameters, the extraction yield of total phenolic content was 6.82 mg/g. An RP-HPLC method for the simultaneous detection of six phenolic compounds (gallic acid, protocatechuic acid, protocatechuic aldehyde, (+)-catechins, chlorogenic acid, and EGCG) was established. The content of phenolic acid in different organs of rattan was investigated. The highest total phenolic content was in the fruit tissues (seeds, episperm, and arillus), followed by the leaves (sheath and blade) and then the stem. Chlorogenic acid was the most abundant phenolic compound in all organs except for the arillus. Finally, the recovery of the phenolic compounds from ChCl-EG extracts by AB-8 macroporous resin was evaluated. The recovery rates of gallic acid, protocatechuic acid, protocatechuic aldehyde, (+)-catechins, chlorogenic acid, and EGCG were more than 90%, and the corresponding purities were 46.57%–67.58%.

## Figures and Tables

**Figure 1 molecules-24-00113-f001:**
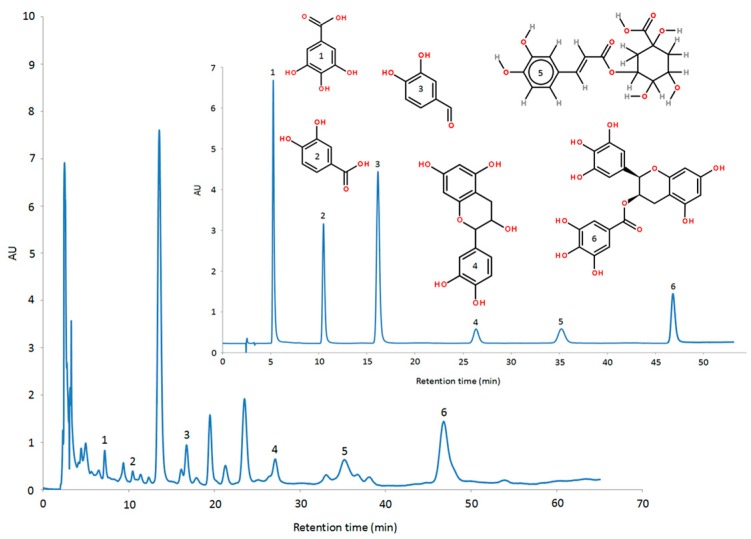
High-performance liquid chromatography (HPLC) of phenolic compounds standards (inner) and Rattan sample (outer).

**Figure 2 molecules-24-00113-f002:**
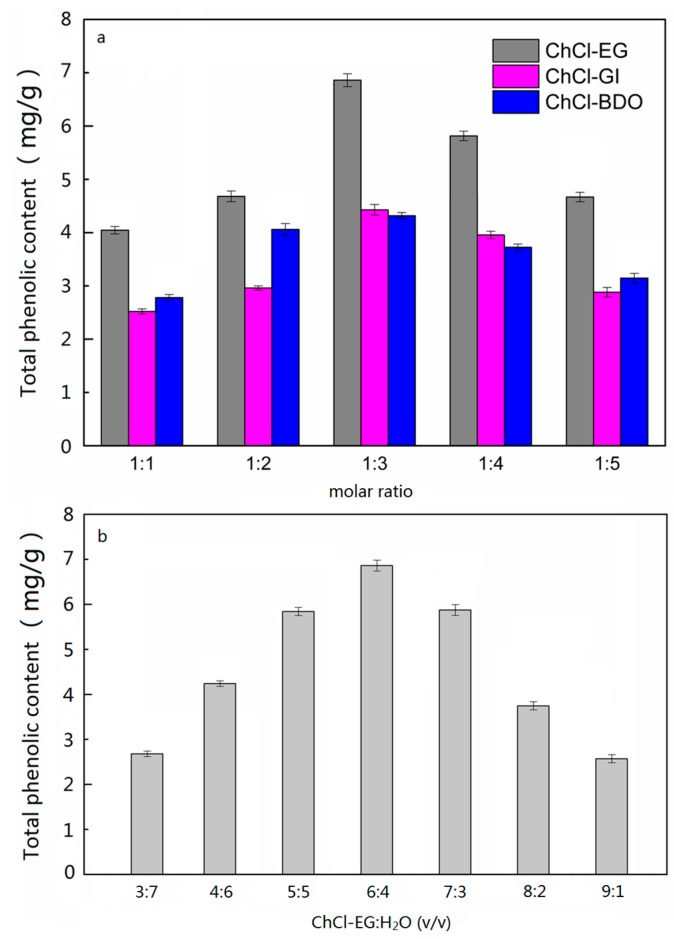
Screening of deep eutectic solvent (DES), including composition of DES (**a**) and amount of water added to DES (**b**).

**Figure 3 molecules-24-00113-f003:**
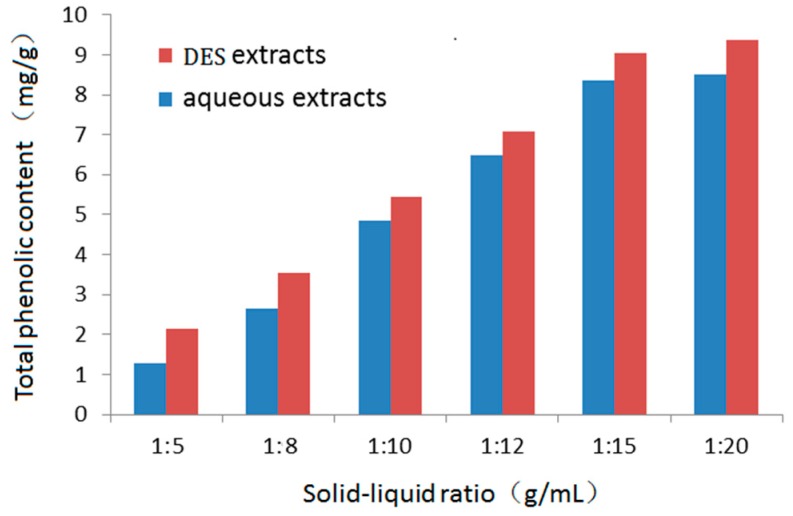
Effect of solid-liquid ratio during homogenate-assisted extraction on total phenolic content.

**Figure 4 molecules-24-00113-f004:**
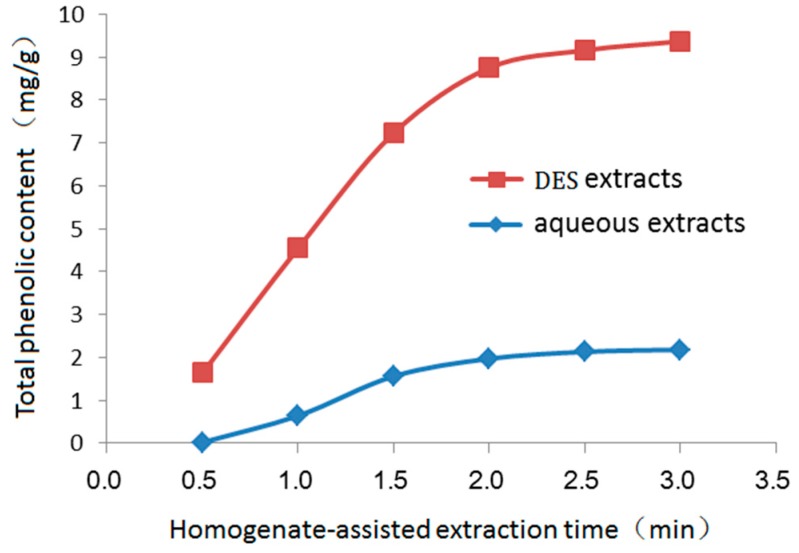
Effect of duration of homogenate-assisted extraction on total phenolic content.

**Figure 5 molecules-24-00113-f005:**
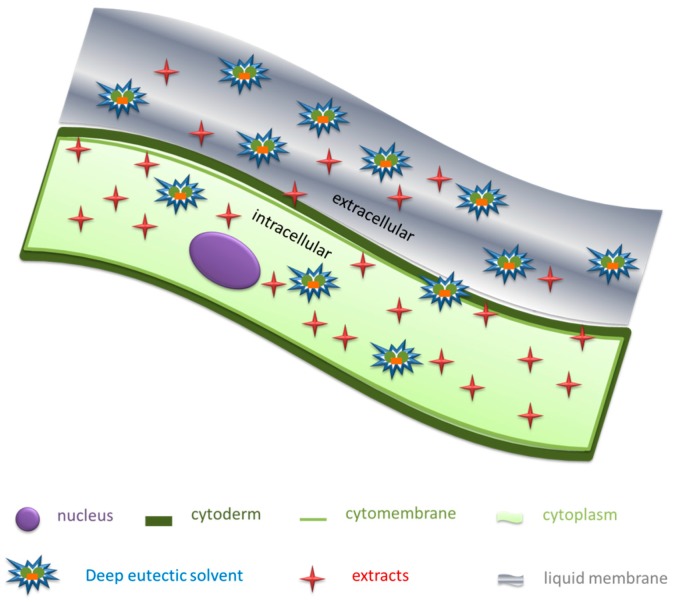
Unstable and non-equilibrium diffusion model of release of target components from plant cells.

**Figure 6 molecules-24-00113-f006:**
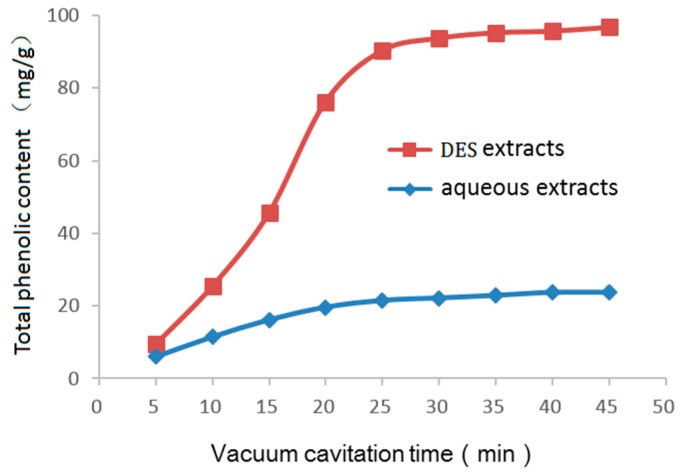
Effect of vacuum-cavitation time on the total phenolic content.

**Figure 7 molecules-24-00113-f007:**
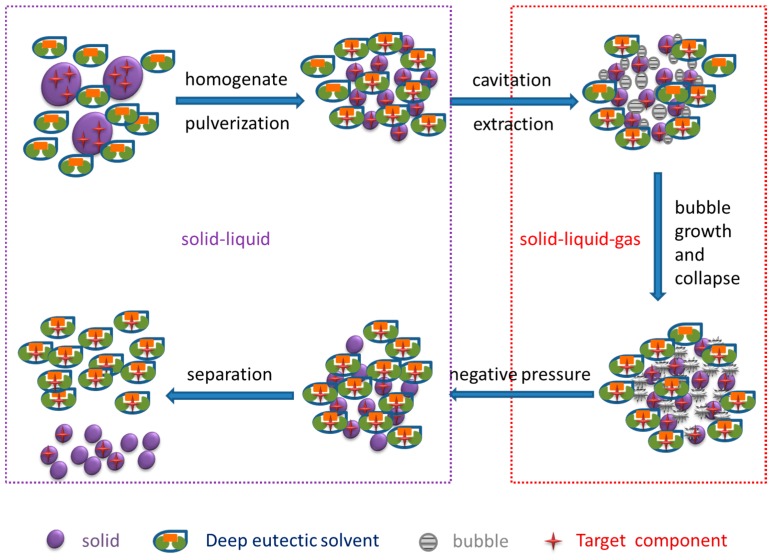
Schematic diagram of homogenate-assisted vacuum-cavitation effect.

**Figure 8 molecules-24-00113-f008:**
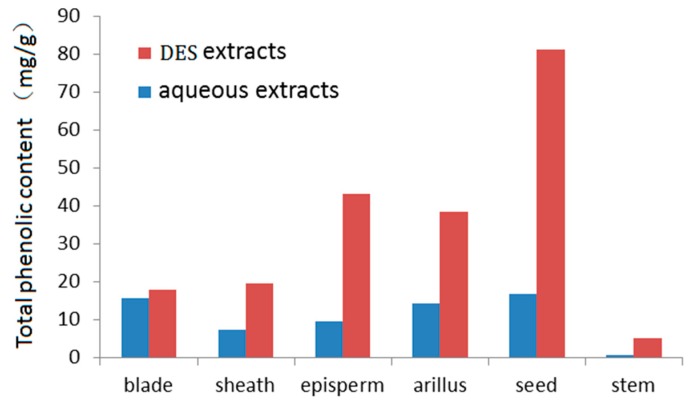
Contents of total phenolic in rattan plant.

**Figure 9 molecules-24-00113-f009:**
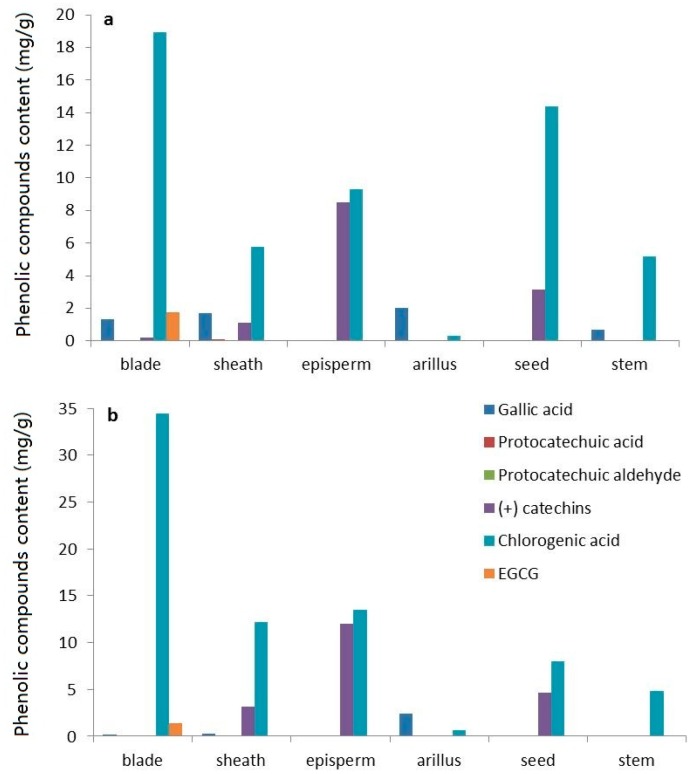
Contents of six phenolic compounds in aqueous extracts (**a**) and DES extracts (**b**) from different parts of rattan plant.

**Table 1 molecules-24-00113-t001:** Calibration curves and limits of detection for six phenolic compounds.

No.	Phenolic Compounds	Retention Time (min)	Corresponding Calibration Curves	*R* ^2^	LOQ (mg/mL)	LOD (μg/mL)
1	Gallic acid	5.2	*Y* = 3.07 × 10^7^*X* + 4.01 × 10^4^	0.9999	0.0157–0.1672	4.33
2	Protocatechuic acid	10.4	*Y* = 5.35 × 10^7^*X* + 4.96 × 10^4^	0.9999	0.0157–0.1676	4.75
3	Protocatechuic aldehyde	16.3	*Y* = 3.10 × 10^7^*X* + 2.34 × 10^4^	0.9999	0.0157–0.1671	4.71
4	(+) catechins	26.7	*Y* = 1.50 × 10^7^*X* + 2.91 × 10^4^	0.9992	0.0156–0.1669	4.65
5	Chlorogenic acid	35.1	*Y* = 2.36 × 10^7^*X* + 9.57 × 10^4^	0.9999	0.0157–0.1676	4.48
6	EGCG	46.8	*Y* = 3.13 × 10^7^*X* + 2.32 × 10^5^	0.9996	0.0157–0.1670	4.92

**Table 2 molecules-24-00113-t002:** Recovery rates of six phenolic compounds by AB-8 resin.

No.	Phenolic Compounds	Upper Flow Rate (mL/min)	Elute Flow Rate (mL/min)	Recovery Rates	Purity (g/g)
1	Gallic acid	0.5	1.0	93.77%	46.57%
2	Protocatechuic acid	0.5	1.0	94.09%	42.51%
3	Protocatechuic aldehyde	0.5	1.0	97.32%	51.94%
4	(+)-catechins	0.5	1.0	97.83%	54.33%
5	Chlorogenic acid	0.5	1.0	94.41%	57.64%
6	EGCG	0.5	1.0	92.47%	67.58%

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
