# Peer review of "Green Extraction of Six Phenolic Compounds from Rattan (Calamoideae faberii) with Deep Eutectic Solvent by Homogenate-Assisted Vacuum-Cavitation Method"

_molecules, 2018, doi:10.3390/molecules24010113_

Round 1

Reviewer 1 Report

This work deals with the homogenate-assisted vacuum-cavitation extraction of phenolic compounds with various deep eutectic solvents  from different parts of rattan.

The experimental design is clear and performed with scientific rigor in order that authors could compare this extraction method with the normal method.

However, some comments will allow to perform the script:

Figure 1 The chromatogram of the standards only shows five compounds, but do not EGCG, and the presence of EGCG is not evident in this chromatogram.

In figure 1, a major peak at ca 12 min appear, but this was not identified, could authors comment about its identity?

Please indicate how the DES solvents used in this work were selected.

Although in the experimental part the different DES employed are defined, in order of clarity, it is necessary to include their definition in the discussion part

Figure 10 is not necessary since Fig 9 displays the same information

In the conclusion, authors claim that the extraction mechanisms of homogenate-assisted negative-pressure cavitation have been analyzed in detail, but they did not perform any experiment related with the study of such mechanism.

Apart to obtain benefit from rattan waste, authors should compare the phenolic content of rattan with other producing phenolic plant species

Author Response

Figure 1 The chromatogram of the standards only shows five compounds, but do not EGCG, and the presence of EGCG is not evident in this chromatogram.

Response: I’m sorry for my mistake, and the correct chromatogram of six standards has been replaced as Figure 1.

In figure 1, a major peak at ca 12 min appear, but this was not identified, could authors comment about its identity?

Response: Yes, we have also noticed this peak at 12 min, but according to our existing standards, this peak was not identified. It may be not a phenolic acid compound. In my further experiment, I'm trying to identify the compound through mass spectrometry and nuclear magnetic resonance. And it will be published in another manuscript. Thanks for reviewer’s constructive comment, in this study, we mainly studied the extraction method with deep eutectic solvents and the content difference of phenolic acid compounds, such as gallic acid, protocatechuic acid, protocatechuic aldehyde, (+)-catechins, chlorogenic acid, and EGCG in different organs.

Please indicate how the DES solvents used in this work were selected.

Response: Through the preliminary experiments, Choline chloride (ChCl) as a kind of hydrogen bond acceptor, polyol as hydrogen bond donor showed a better extraction efficiency of phenolic acid compounds. Thus, in this study, the polyol (ethylene glycol (EG), glycerol (GI) and 1,4-butanediol (BDO)) as hydrogen bond donor was screened. And the extraction yield of the total phenol content as the response value, the different DESs (ChCl-EG, ChCl-GI, ChCl-BDO) was investigated. The results were showed in Figure 2. This explanation was added in section 2.3.1.

Although in the experimental part the different DES employed are defined, in order of clarity, it is necessary to include their definition in the discussion part

Response: Thanks for reviewer’s constructive comment, and the different DES employed are defined in section 2.3.1 was added.

Figure 10 is not necessary since Fig 9 displays the same information

Response: Figure 10 and section 2.8 have been deleted. But the contents of six phenolic compounds in aqueous extracts and in DES extracts from different parts of rattan plant were showed in Fig.9. And in Fig.10, from the point of view of a single phenolic acid, a distribution percentage was investigated. Although the data is repeated, the expression angle is different. It was clear described which organ has the highest content of a single phenolic compound (distribution percentage was marked by different colours). If Fig.10 (and section 2.8) was necessary, they will be added in the text again.

In the conclusion, authors claim that the extraction mechanisms of homogenate-assisted negative-pressure cavitation have been analyzed in detail, but they did not perform any experiment related with the study of such mechanism.

Response: The extraction mechanisms of homogenate-assisted negative-pressure cavitation have been analyzed according to our previous studies. ([31] Yinnan Sun , Kui Yang , Qin Cao , Jinde Sun , Yu Xia , Yinhang Wang, Wei Li, Chunhui Ma*, Shouxin Liu, Homogenate-assisted vacuum-powered bubble extraction of Moso bamboo flavonoids for on-line scavenging free radical capacity analysis. Molecules, 2017, 21, 204-217. [32] Yu Xia, Yinhang Wang, Wei Li, Chunhui Ma*, Shouxin Liu, Homogenization-assisted cavitation hybrid rotation extraction and macroporous resin enrichment of dihydroquercetin from Larix gmelinii. Journal of Chromatography B, 2017, 82, 204-217.) The study of the mechanism has been very mature, but the extraction mechanism with deep eutectic solvent is explained in this text for the first time. It is explained the formation of liquid film on the plant material surface, from a microscopic point of view (the plant cell and deep eutectic solvent). The innovation lies in the formation of hydrogen bonds that decrease the barrier of plant cell walls recalcitrance. And this effect was worked synergistically with homogenization extraction and negative-pressure cavitation extraction. The references and explanation were added in 2.4.3 and 2.5.2.

Apart to obtain benefit from rattan waste, authors should compare the phenolic content of rattan with other producing phenolic plant species

Response: Thanks for reviewer’s constructive comment, previous research has focused on the commercial aspects of rattan as a raw material for furniture and handicrafts, rather than its chemical composition or biological activity, and thus there is not much relevant literature. In other experiments on the production of phenolic plant species, Yu et al. used ultrasonic-assisted extraction to extract phenolic substances from peanut shells with an optimum yield of up to 64.5 mg/g; Li et al. found that the methanol extract of S. serrata has high antioxidant activity and the total phenolic mass fraction is 52.35 mg/g. It has been found that the vines used in this study not only utilize waste resources, but also extract more phenolic compounds for use as active ingredients. In our study, the total phenolic content reached to 96.47 ± 2.02 mg/g with DES by homogenate-assisted negative-pressure cavitation extraction.

Reviewer 2 Report

This paper describes the homogenate-assisted vacuum-cavitation extraction (HVE) method using deep eutectic solvents (DES). The article is well written, it is of interest to the scientific community and the methods used are appropriate. The work is interesting and deepens in the knowledge on the "green" extraction of important bioactive phenolic compounds from rattan (Calamoideae faberii).

Three DESs with different compositions, including ethylene glycol (ChCl-EG), glycerol (ChCl-Gl), and 1, 4-butylene glycol (ChCl-BDO) were prepared using three hydrogen bond donors (HBD). A DES extraction method was combined homogenate technology with vacuum-cavitation for the use acceleration of the extraction of phenolic compounds from rattan waste materials. The extraction mechanism of homogenate-assisted vacuum-cavitation was compared with that of ultrasonic cavitation. In addition, a reversed-phase high-performance liquid chromatography (RP-HPLC) was used to quantitative determination of six phenolic compounds distributed in each extract and in different parts of the rattan plant.

However, this work contains several drawbacks existed in the manuscript and modification of the manuscript is needed for publication in the Molecules. I believe that this work will be better with major revisions.

1.       Abstract

Pg. 1: The sentence “The highest total phenolic… (sheath and blade)”: Please add the amount detected of total phenolic content in the plant tissues and each of six phenolic compounds.

2.       Introduction

Pg. 2:  The penultimate section in the chapter Introduction "Phenolic compounds can...practical opportunities." is redundant.

3.       Results and Discussion

Pg. 3: Table 1: Add the LOQ parameter and Linearity range of the phenolics analysis.

Pg. 4: in the Subsection 3.3.1. Add a full name for ChCl-EG, ChCl-Gl, and ChCl-BDO in the sentence “In Fig. 2… are displayed”.

Pg. 4: first sentence in the Subsection 3.3.2.:  Does the viscosity of DES depend on the temperature and how did you select the temperature for the DES preparation in this paper? In addition, it would be useful to show a scheme of a eutectic point on a two-component phase diagram used in this paper (ChCl-EG, ChCl-Gl, and ChCl-BDO) as Supplementary Material to present properties of the used DESs.

Pg. 4: Subsection 3.3.2.: How can it explain that the greatest phenolic content in extractions where is donor EG (Fig.2a)?

Pg. 4: Subsection 3.3.2.: the sentence “However, excessive water addition… the polarity of the mixture” - how did the authors determine differences in the polarity of the mixture? What is the polarity of the chosen volume ratio of ChCl-EG: H2O = 6: 4 compared to the others?

Pg. 4.: Section 2.4. The results are confusingly described to find optimal extraction conditions.

-          Subsection 2.4.1. The solid-liquid ratio describes DES homogenate-assisted extraction at different solid-liquid ratios. However, in the Fig.3 is presented total phenolic content of DES extract and ethanolic extract.

-          In the selection of the duration of extraction (subsection 2.4.2) at the same selected ratio from the previous subsection (2.4.1), the duration of the extraction was selected, however, based on the extraction with 80% methanol or pure water as a solvent, and in Fig.4 is presented total phenolic content in DES and aqueous extract to choose optimal extraction time (Fig.4).

Due to this confusion, it is suggested the authors prepare a Table with working and chosen optimal conditions for homogenate-assisted extraction (add in Supplementary Material).

Pg. 6: subsection 2.4.3. - How did the authors test the effect of the size of raw material in homogenate-assisted extraction?

Pg. 7: Section 2.5. Factors affecting cavitation extractions - What source of cavitation was used? Ultrasonic bath or? Which equipment?

Pg. 7: Subsection 2.5.1. - The comparison between results obtained with homogenate-assisted extraction and ultrasonic-assisted extraction (subsections 2.4.2. and 2.5.1) missing.

Pg. 7: Subsection 2.5.2. - Did the authors measure the pressure during extraction? Figure 7 should not be in the section Results and Discussion. It is more appropriate for the introductory part.

Pg. 8-9: I suggest a new Table with Results of total phenolic content and phenolic compound contents among different parts of rattan (instead of Figure 8 and 9). What is the difference between Figure 8 and 9? How was extracted phenolics showed in Figure 9 described as b) from different parts of the rattan plant (Pg. 10)?

Pg. 10: the title of subsection 2.8 is the Content distribution of individual phenolic compounds in aqueous and ethanolic extracts among different, however, in Figure 10a,c,e,g,I was presented the amount of individual phenolic in DES extracts, and in 10b,d,f,h,j was showed the amount of individual phenolic in aqueous extracts (Pg. 11). Please, explain.

Pg. 12: Why ethanol extract was used for the separation of individual phenolic compounds be chosen? Did the authors use only one adsorbent?

Therefore, I propose that in Section 3. Materials and Methods add the scheme of experiments with an explanation of their purpose. This will make the work more clear and legible for readers.

Additionally:

1.       The Results are described, but with very poor Discussion.

2.       Furthermore, the Graphical abstract and Line numbers in the manuscript are missing.

Conclusion:

This manuscript can be accepted after the implementation of suggested changes and corrections.

Author Response

1.       Abstract

Pg. 1: The sentence “The highest total phenolic… (sheath and blade)”: Please add the amount detected of total phenolic content in the plant tissues and each of six phenolic compounds.

Response: The total phenolic content in different plant tissues has been added in Abstract.

2.       Introduction

Pg. 2:  The penultimate section in the chapter Introduction "Phenolic compounds can...practical opportunities." is redundant.

Response: Yes, it has been removed.

3.       Results and Discussion

Pg. 3: Table 1: Add the LOQ parameter and Linearity range of the phenolics analysis.

Response: Thanks for reviewer’s constructive comment, the LOQ parameter and LOD of the phenolics analysis was added in Table 1.

Pg. 4: in the Subsection 3.3.1. Add a full name for ChCl-EG, ChCl-Gl, and ChCl-BDO in the sentence “In Fig. 2… are displayed”.

Response: Yes, the full name for ChCl-EG, ChCl-Gl, and ChCl-BDO was added in section 2.3.1, the original 3.3.1.

Pg. 4: first sentence in the Subsection 3.3.2.:  Does the viscosity of DES depend on the temperature and how did you select the temperature for the DES preparation in this paper? In addition, it would be useful to show a scheme of a eutectic point on a two-component phase diagram used in this paper (ChCl-EG, ChCl-Gl, and ChCl-BDO) as Supplementary Material to present properties of the used DESs.

Response: DESs were synthesized by a heating method [36]. The hydrogen bond acceptor in each DES was ChCl. This was mixed with a hydrogen bond donor (one of the polyols listed in 2.3.1) in an appropriate molar ratio and placed in a reaction flask with magnetic agitation at 80 °C for 30 min, resulting in the formation of a homogeneous liquid. The melting point of ChCl is 302-305°C, and the melting point of EG, GL, and BDO were -13°C, 18.6°C, and 20.2°C, respectively. A eutectic point on a two-component phase was far lower than the melting point of each component (80°C was far lower than 302-305°C).

The viscosity of DES depends on the temperature. However, the temperature for homogenate-assisted vacuum-cavitation extraction was operated at room temperature. The viscosity of DES solution depends on the water content in DES solution. Therefore, in our study, we chose to change the water content of DES to increase its osmotic effect, enhancing the mass transfer from plant matrices to solution, and thus affecting the extraction ability of the target compound. The viscosity of DES solution was investigated as Supplementary Material (Fig.2S). When the water content lower than 30%(v/v),the viscosity of DES solution was increased sharply.

Fig. 2S Effect of water content on the viscosity of DES solution

Pg. 4: Subsection 3.3.2.: How can it explain that the greatest phenolic content in extractions where is donor EG (Fig.2a)?

Response: In Fig. 2a, the extraction yields when using ChCl-EG, ChCl-Gl, and ChCl-BDO with different molar ratios (1:1, 1:2, 1:3, 1:4, 1:5) are displayed. In all three DESs, hydrogen bonds are shared between the HBD (polyol) and chloride ions from ChCl (HBA), as showed in Fig.1S, for polyols, there are more than two hydroxyl groups in the molecule that can form large molecules by forming intramolecular hydrogen bonds, and can also form intermolecular hydrogen bonds with ChCl. The more number of OH was provide by GI, when the same molar ratios of polyol, but the better water absorbent performance of GI may form the intermolecular hydrogen bonds, resulting in a decrease in the number of hydrogen bonds with ChCl. Compared with BDO, the steric effects to the formation of intermolecular hydrogen bonds with is ChCl smaller. Therefore, the extraction yield of total phenolic compounds with ChCl-EG (6.97 ± 0.11 mg/g) was higher than that with ChCl-Gl (4.79 ± 0.13 mg/g) and ChCl-BDO (4.77 ± 0.14 mg/g), and 1:3 was the optimum molar ratio of ChCl and polyol. This explanation was added in section 2.3.1.

Fig. 1S Formation diagram of hydrogen bonds between ChCl and EG

Pg. 4: Subsection 3.3.2.: the sentence “However, excessive water addition… the polarity of the mixture” - how did the authors determine differences in the polarity of the mixture? What is the polarity of the chosen volume ratio of ChCl-EG: H2O = 6: 4 compared to the others?

Response: I’m sorry for the ambiguous sentences and the sentence was revised to “However, excessive water addition can increase the chemical polarity of DES solution, and the dissolve of phenolic compounds decreased according to the “similar solubility principle”. Therefore, the volume ratio of ChCl-EG:H2O = 6:4 was used for further extraction processes.” in section 3.3.2.

Pg. 4.: Section 2.4. The results are confusingly described to find optimal extraction conditions.

Response: The optimal extraction conditions for homogenate extraction were revised in section 2.4.2.

-Subsection 2.4.1. The solid-liquid ratio describes DES homogenate-assisted extraction at different solid-liquid ratios. However, in the Fig.3 is presented total phenolic content of DES extract and ethanolic extract.

Response: I’m sorry for my mistake, and the Fig.3 was replaced.

-In the selection of the duration of extraction (subsection 2.4.2) at the same selected ratio from the previous subsection (2.4.1), the duration of the extraction was selected, however, based on the extraction with 80% methanol or pure water as a solvent, and in Fig.4 is presented total phenolic content in DES and aqueous extract to choose optimal extraction time (Fig.4).

Due to this confusion, it is suggested the authors prepare a Table with working and chosen optimal conditions for homogenate-assisted extraction (add in Supplementary Material).

Response: I’m sorry for my mistake, 80% methanol as a compared solvent was used for homogenate extraction. But the data was not showed in this manuscript. Thanks for reviewer’s constructive comment, and the optimal conditions for homogenate-assisted extraction were listed in Table 1S as Supplementary Material.

Table 1S Parameters of homogenate-assisted extraction

No.

Operation

solid-liquid ratio (g/mL)

duration time (min)

solvent

total phenolic content (mg/g)

1-1

Homogenate

1:15

2.0

DES

8.75 ± 0.16

1-2

Homogenate

1:15

2.0

Pure water

1.97 ± 0.04

1-3

Homogenate

1:15

2.0

80% methanol

6.44 ±   0.11

2-1

vacuum-cavitation

1:15

25.0

DES

90.33 ± 1.89

2-2

vacuum-cavitation

1:15

25.0

Pure water

21.44 ± 0.44

2-3

vacuum-cavitation

1:15

25.0

80% methanol

74.55 ± 1.23

3-1

ultrasonic-assisted

1:15

25.0

DES

84.32 ± 1.98

3-2

ultrasonic-assisted

1:15

25.0

Pure water

17.25 ± 0.64

3-3

ultrasonic-assisted

1:15

25.0

80% methanol

64.83 ± 1.14

Pg. 6: subsection 2.4.3. - How did the authors test the effect of the size of raw material in homogenate-assisted extraction?

Response: In homogenate-assisted extraction process, shearing of solid particles into smaller pieces, the particle size of raw material is getting smaller and smaller with the homogenate time increasing. The shearing action reduces the size of the large solid particles and destroys most of the plant cell walls. At the same time, a liquid membrane layer which contained more hydrogen bonds forms at the cell surface, the permeability of cell was increased and the cell contents phenolic acids are released into the solvent. This explanation was added in section 2.4.3.

Pg. 7: Section 2.5. Factors affecting cavitation extractions - What source of cavitation was used? Ultrasonic bath or? Which equipment?

Response: The generation of vacuum-cavitation by negative pressure is a cheap and energy-efficient method. As shown in Fig. 6, extractions were carried out at negative pressure with a 180 W vacuum pump for 5, 10, 15, 20, 25, 30, 35, 40, and 45 min. The sentence “The method can maintain a constant low temperature with an intensity as strong as that of ultrasonic cavitation.” was deleted.

Pg. 7: Subsection 2.5.1. - The comparison between results obtained with homogenate-assisted extraction and ultrasonic-assisted extraction (subsections 2.4.2. and 2.5.1) missing.

Response: In our experiment, ultrasonic-assisted extraction as a comparison method was operated. However, in this manuscript, the subject of study is vacuum-cavitation method with DES. So, the extraction yields of phenolic acids by ultrasonic-assisted extraction were not showed in the text. The data was added in Table 1S.

Pg. 7: Subsection 2.5.2. - Did the authors measure the pressure during extraction? Figure 7 should not be in the section Results and Discussion. It is more appropriate for the introductory part.

Response: Extractions were carried out at negative pressure with a 180 W vacuum pump, and the maximum vacuum can reached to -0.1Mpa (in section 3.2.7). From Figure 7, the negative-pressure cavitation effect was interpreted according to our previous studies. ([31] Yinnan Sun , Kui Yang , Qin Cao , Jinde Sun , Yu Xia , Yinhang Wang, Wei Li, Chunhui Ma*, Shouxin Liu, Homogenate-assisted vacuum-powered bubble extraction of Moso bamboo flavonoids for on-line scavenging free radical capacity analysis. Molecules, 2017, 21, 204-217. [32] Yu Xia, Yinhang Wang, Wei Li, Chunhui Ma*, Shouxin Liu, Homogenization-assisted cavitation hybrid rotation extraction and macroporous resin enrichment of dihydroquercetin from Larix gmelinii. Journal of Chromatography B, 2017, 82, 204-217.) The study of the mechanism has been very mature, but the extraction mechanism with deep eutectic solvent is explained in this text for the first time. It is explained the formation of liquid film on the plant material surface, from a microscopic point of view (the plant cell and deep eutectic solvent). The innovation lies in the formation of hydrogen bonds that decrease the barrier of plant cell walls recalcitrance. And this effect was worked synergistically with homogenization extraction and negative-pressure cavitation extraction. The references and explanation were added in 2.4.3 and 2.5.2.

Pg. 8-9: I suggest a new Table with Results of total phenolic content and phenolic compound contents among different parts of rattan (instead of Figure 8 and 9). What is the difference between Figure 8 and 9? How was extracted phenolics showed in Figure 9 described as b) from different parts of the rattan plant (Pg. 10)?

Response: The distribution of total phenolic content in different organs was described in Figure 8, while the distribution of 6 specific phenolic compounds was described in Figure 9.

Pg. 10: the title of subsection 2.8 is the Content distribution of individual phenolic compounds in aqueous and ethanolic extracts among different, however, in Figure 10a,c,e,g,I was presented the amount of individual phenolic in DES extracts, and in 10b,d,f,h,j was showed the amount of individual phenolic in aqueous extracts (Pg. 11). Please, explain.

Response: Figure 10 and section 2.8 have been deleted. But the contents of six phenolic compounds in aqueous extracts and in DES extracts from different parts of rattan plant were showed in Fig.9. And in Fig.10, from the point of view of a single phenolic acid, a distribution percentage was investigated. Although the data is repeated, the expression angle is different. It was clear described which organ has the highest content of a single phenolic compound (distribution percentage was marked by different colours). If Fig.10 (and section 2.8) was necessary, they will be added in the text again. The data in Figure 10a,c,e,g,I was obtained by extracting with DES, and the data in 10b,d,f,h,j was obtained by extracting with pure water.

Pg. 12: Why ethanol extract was used for the separation of individual phenolic compounds be chosen? Did the authors use only one adsorbent?

Response: I’m sorry for my mistake, 80% methanol as a compared solvent was used for homogenate-assisted negative-pressure cavitation extraction. But the data was not showed in this manuscript. The data were listed in Table 1S as Supplementary Material.

Therefore, I propose that in Section 3. Materials and Methods add the scheme of experiments with an explanation of their purpose. This will make the work more clear and legible for readers.

Response: The scheme of experiments with an explanation of their purpose was according to our previous studies. ([31] Yinnan Sun , Kui Yang , Qin Cao , Jinde Sun , Yu Xia , Yinhang Wang, Wei Li, Chunhui Ma*, Shouxin Liu, Homogenate-assisted vacuum-powered bubble extraction of Moso bamboo flavonoids for on-line scavenging free radical capacity analysis. Molecules, 2017, 21, 204-217.). Thanks for reviewer’s constructive comment, and the reference was added in section 3.2.5.

Additionally:

1. The Results are described, but with very poor Discussion.

Response: We added some discussion try our best in the text.

2. Furthermore, the Graphical abstract and Line numbers in the manuscript are missing.

Response: The Graphical abstract and Line numbers have been added.

Reviewer 3 Report

This work deals with the evaluation of specific phenolic compounds extracted by means of Deep Eutectic Solvents (DES) and homogenate-assisted vacuum-cavitation process from rattan byproducts. This research topic is very interesting, because it is possible to provide greener extraction processes in order to improve the bioactive molecules yielding. In general, the document is well written, and the discussion seems to be appropriate to results information. Nevertheless, there are some specific issues and questions for authors that require to be attended before to accept this manuscript for publication, since some sections results unclear and confusing.

 - In general terms, the manuscript presents several aims to be achieved and I consider that they can be grouped as: the comparison of DES solvent extraction against conventional solvents like water, the presence of water in DES, ethanol or methanol for stablishing the best extraction solvent; the effect of cavitation with and without ultrasound for better transport phenomena and therefore higher phenolic compounds contents and their comparison against conventional maceration for extraction process; the relevance of the type of plant tissue for phenolic compounds recovery with the combination of the extraction conditions stated before. All of them are important, but authors should center clearly the main objective of the work, in terms of which is more important in agreement to the title, the type of solvent, the extraction technique, the plant tissue, phenomenological issues, cavitation with or without vacuum assisted.

-  Also, it is highly recommended to avoid duplicate information in the complete document, this means that if certain values were presented within a Table or a Figure, authors should avoid describing it in the text, repeating the same information.

-  Section 2.1. It is important to define that “aqueous extracts” are those extracts made with pure water, since could create confusions when ethanolic or methanolic aqueous solutions are used for extraction solvents. This is very important, because in later sections, some aqueous extractions are compared with DES extracts, but they include or do not the alcoholic samples.

-  It is stated “The content of phenolic compounds in the aqueous extract (extracted by heating and stirring with water for 6 h….” Indicate the temperature used for these extractions, and how does it affect the phenolics extraction compared to the rest of extraction methods that were carried out at room temperature. How do authors ensure that after 6 hours of extraction the ideal extraction time was achieved? How do long times at high temperature affect the phenolic contents?

- The same questions for the aqueous ethanol (80% v/v?) extraction solvent, where extraction was carried out by Soxhlet. Why do authors consider that 6 h for extraction process results in appropriate condition for the type of bioactive molecules to be yielded?

- Figure 1. In the chromatogram shown, besides the identified phenolic compounds there are at least five signals with larger peak areas that those for standard compounds, why do authors did not consider presenting an HPLC-MS for identifying them? Because probably they have a significant contribution to the total phenolic contents and therefore to biological activities.

- Sección 2.3. Numbering of sections 2.3.1 and 2.3.2 are wrong, they appear like 3.3.1 and 3.3.2 respectively.

 - It is stated “The extraction yield of total phenolic compounds with ChCl-EG (6.97 mg/g) was higher than that with ChCl-Gl and ChCl-BDO, and 1:3 was the optimum molar ratio of ChCl and polyol”. First, indicate if the mass ratio reported is in dry or wet basis, and specify if there are some mg of phenolics per gram of what. Second, how do authors ensure that 1:3 molar ratio represents the “optimum condition”, if there is not shown any statistics that supports it? Finally, include the statistical analysis for establish that ChCl-Gl resulted in the significant best extraction solvent.

 - Figure 2a.  Indicate the water content used for this figure; when it is compared to Figure 2b at different water amounts added to DES, the higher phenolic compounds extraction in both figures seems to be the same value, but the effect described in the text about improving the transport phenomena and therefore the extraction yielding is not clear at all.

Figure 3. Is there significant differences in the solid-liquid ratios between solvents for extraction? Please add standard deviation to data in tables and figures.

- Section 2.4.1. Are these extraction conditions (time and phenolic content) comparable to those presented in section 2.1? Why did authors not homogenize the extraction times used in all the extraction processes?

Section 2.4.2. It is not clear why do authors use aqueous ethanol or methanol indistinctly for establishing different extraction conditions (times, water content, solid:liquid ratio, etc.) and then select them as the best extraction conditions because they resulted in higher extract contents, without considering the effect of solvent polarity and its effect of the phenolics solubility. There are several works in literature that have demonstrated that methanol shows significant differences when compared to ethanol or water solvents for phenolic compounds extraction. It is recommended to homogenize the solvents and matrices used for the establishment of the “best” or “optimum” extraction conditions.

-                    Figure 4. Discussion in section 2.4.2 only indicate the effect of aqueous methanol or water on the total phenolic content, but in Figure 4 appears two profiles corresponding to aqueous extracts and DES extracts. The discussion about DES extraction is not contained in this section. In agreement to section 2.1 where 6 h were used for extraction, why was a maximum of 3 min used for the rest of the extraction methods? Then, results in section 2.1 can be comparable to those shown in the rest of the document?

- Section 2.4.3. It is stated “During homogenization process, the thickness ….” What homogenization conditions were used in this section? They were not specified in M&M section.

- Figure 5. Please add the transport phenomena signaling and improve the transport phenomena discussion, indicate if under the extraction conditions described if the diffusion mechanism or convective resistance to mass transfer results in the most important phenomena that control the phenolic compounds extraction.

- Section 2.5. Authors presents a lot of experiments, I consider that a good design of experiments could be very helpful for elucidating the most important parameters and variables that affect the phenolic content in extracts (solvent, time, cavitation, vacuum, etc).

-                    Section 2.5.2. This section is merely descriptive about cavitation phenomena, it does not include results information, it could be incorporated in another section that already includes results data.

-                    Section 2.6. It should be convenient to indicate which extraction conditions were stated by authors as the “optimum extraction conditions”. Please do not repeat information within figures and main text. What is the relevance for this work when authors differentiate the phenolic contents in the diverse plant tissues? In agreement to introduction section, most of these tissues are discarded as byproducts, why do authors not only blend all the plant wastes and just focused this document for defining the use of DES and non-conventional extraction process for phenolics yield from rattan plant?

- Section 2.7. It is recommended to show the chromatograms obtained from different plant tissues, in order to show if all the tissues present the same trend of phenolics or there is a dependence of the type of phenolics with the tissues where they were extracted.

- Section 2.8. There is a misunderstood between the title of the section and the figure that shows the results information, this means that title includes the analysis of aqueous and ethanolic extracts, but figure 10 only includes DES and aqueous extracts, and at the end it is not clear what extraction conditions were used at this point.

-   Section 3.1.1. Indicate if the whole rattan plant was purchase in the same place, time, supplier, etc. Indicate the year of the purchase.

Table 2. Information in this table is already described in the text.

- Section 3.2.1. indicate if a gallic acid standard curve was used.

- Section 3.2.3. indicate the mixing time for DES preparation and which was the criteria used for establishing the DES formation.

- Section 3.2.4, Temperature of extraction is not declared, does it affect the extraction yield? What was the particle size and polydispersity in plant material achieved under this homogenization process? Was it the same for all the plant tissues? Could particle size differences affect the phenolics extraction?

- In M&M section is not clear what procedures were done with ethanol, methanol or water like solvents in order to establish the optimum conditions for extraction. Also, how do authors ensure the finding of “optimum extraction conditions” if there is not any statistical analysis done?

- It is highly recommended to use specialized statistical software for a more rigorous analysis like means comparisons tests, surface response methodology, design of experiments.

- Section 4. In agreement to chromatograms, there are at least 4 peaks signals with higher areas than the standard compounds used, it is recommended to make the identification of these compounds since they may affect in bigger proportion the effect on phenolics quantification.

- Because formation of DES depends on equilibrium relationships, how does the water addition affect such equilibrium and therefore their functionality for improving the phenolics recovery?

-  Indicate standard deviation values for quantities described along the complete document.

It seems to be a very broad document with a lot of important information, its handle sometimes could result in a very complicated task, where all the activities should be interrelated among each other and provide trusty information. In this sense, it is recommended to authors to be clearer in the presentation, discussion and interrelationship of information, selecting the mains effects for achieving all the objectives raised.

Author Response

 - In general terms, the manuscript presents several aims to be achieved and I consider that they can be grouped as: the comparison of DES solvent extraction against conventional solvents like water, the presence of water in DES, ethanol or methanol for stablishing the best extraction solvent; the effect of cavitation with and without ultrasound for better transport phenomena and therefore higher phenolic compounds contents and their comparison against conventional maceration for extraction process; the relevance of the type of plant tissue for phenolic compounds recovery with the combination of the extraction conditions stated before. All of them are important, but authors should center clearly the main objective of the work, in terms of which is more important in agreement to the title, the type of solvent, the extraction technique, the plant tissue, phenomenological issues, cavitation with or without vacuum assisted.

Response: Thanks for reviewer’s constructive comment, and the relevant details have been modified to highlight the main objectives of the study.

-  Also, it is highly recommended to avoid duplicate information in the complete document, this means that if certain values were presented within a Table or a Figure, authors should avoid describing it in the text, repeating the same information.

Response: Yes, we checked the whole text, and the duplicate information in the text has been deleted.

-  Section 2.1. It is important to define that “aqueous extracts” are those extracts made with pure water, since could create confusions when ethanolic or methanolic aqueous solutions are used for extraction solvents. This is very important, because in later sections, some aqueous extractions are compared with DES extracts, but they include or do not the alcoholic samples.

Response: I’m sorry for my mistake, 80% methanol as a compared solvent was used for homogenate extraction. But the data which was listed in Table 1S as Supplementary Material was not showed in this manuscript. The data listed in the text was all from “DES extracts” and “aqueous extracts”, and the "aqueous extracts" have been defined as extracted with pure water in section 2.1.

Table 1S Parameters of homogenate-assisted extraction

No.

Operation

solid-liquid ratio (g/mL)

duration time (min)

solvent

total phenolic content (mg/g)

1-1

Homogenate

1:15

2.0

DES

8.75 ± 0.16

1-2

Homogenate

1:15

2.0

Pure water

1.97 ± 0.04

1-3

Homogenate

1:15

2.0

80% methanol

6.44 ±   0.11

2-1

vacuum-cavitation

1:15

25.0

DES

90.33 ± 1.89

2-2

vacuum-cavitation

1:15

25.0

Pure water

21.44 ± 0.44

2-3

vacuum-cavitation

1:15

25.0

80% methanol

74.55 ± 1.23

3-1

ultrasonic-assisted

1:15

25.0

DES

84.32 ± 1.98

3-2

ultrasonic-assisted

1:15

25.0

Pure water

17.25 ± 0.64

3-3

ultrasonic-assisted

1:15

25.0

80% methanol

64.83 ± 1.14

-  It is stated “The content of phenolic compounds in the aqueous extract (extracted by heating and stirring with water for 6 h….” Indicate the temperature used for these extractions, and how does it affect the phenolics extraction compared to the rest of extraction methods that were carried out at room temperature. How do authors ensure that after 6 hours of extraction the ideal extraction time was achieved? How do long times at high temperature affect the phenolic contents?

Response: In order to analysis the chemical composition of rattan materials, “aqueous extracts” and “ethanolic extracts” were investigated. It takes longer extraction time to make sure the extraction is complete. Moreover, the mass of “aqueous extracts” as a compared data, compared with the total phenolic content in “aqueous extracts”. There are some other compounds (except phenolic compounds) in “aqueous extracts”.

- The same questions for the aqueous ethanol (80% v/v?) extraction solvent, where extraction was carried out by Soxhlet. Why do authors consider that 6 h for extraction process results in appropriate condition for the type of bioactive molecules to be yielded?

Response: In order to analysis the chemical composition of rattan materials, “aqueous extracts” and “ethanolic extracts” were investigated. It takes longer extraction time to make sure the extraction is complete. The selection of the 6-hour extraction time was obtained by reviewing some of the literature, as it was not the focus of this study and therefore was not specifically optimized.

Figure 1. In the chromatogram shown, besides the identified phenolic compounds there are at least five signals with larger peak areas that those for standard compounds, why do authors did not consider presenting an HPLC-MS for identifying them? Because probably they have a significant contribution to the total phenolic contents and therefore to biological activities.

Response: Yes, we have also noticed these peaks, but according to our existing standards (common phenolic acid), this peak was not identified. It may be not a phenolic acid compound. In my further experiment, I'm trying to identify the compound through mass spectrometry and nuclear magnetic resonance. And it will be published in another manuscript. Thanks for reviewer’s constructive comment, in this study, we mainly studied the extraction method with deep eutectic solvents and the content difference of phenolic acid compounds, such as gallic acid, protocatechuic acid, protocatechuic aldehyde, (+)-catechins, chlorogenic acid, and EGCG in different organs.

Section 2.3. Numbering of sections 2.3.1 and 2.3.2 are wrong, they appear like 3.3.1 and 3.3.2 respectively.

Response: I’m sorry for my mistake, the numbering of sections “3.3.1” and “3.3.2” has been revised to “2.3.1” and “2.3.2”.

 - It is stated “The extraction yield of total phenolic compounds with ChCl-EG (6.97 mg/g) was higher than that with ChCl-Gl and ChCl-BDO, and 1:3 was the optimum molar ratio of ChCl and polyol”. First, indicate if the mass ratio reported is in dry or wet basis, and specify if there are some mg of phenolics per gram of what. Second, how do authors ensure that 1:3 molar ratio represents the “optimum condition”, if there is not shown any statistics that supports it? Finally, include the statistical analysis for establish that ChCl-Gl resulted in the significant best extraction solvent.

Response: It’s not “mass ratio”. In Fig. 2a, the extraction yields when using ChCl-EG, ChCl-Gl, and ChCl-BDO with different molar ratios (1:1, 1:2, 1:3, 1:4, 1:5) are displayed. The extraction yield of total phenolic compounds with ChCl-EG (6.97 ± 0.11 mg/g) was higher than that with ChCl-Gl (4.79 ± 0.13 mg/g) and ChCl-BDO (4.77 ± 0.14 mg/g). So ChCl-EG is selected extraction solvent for future experiment, and 1:3 was the optimum molar ratio of ChCl and EG from the following Fig. 2a.

Figure 2 Screening of the composition of DES

 - Figure 2a.  Indicate the water content used for this figure; when it is compared to Figure 2b at different water amounts added to DES, the higher phenolic compounds extraction in both figures seems to be the same value, but the effect described in the text about improving the transport phenomena and therefore the extraction yielding is not clear at all.

Response: In Figure 2(a) screening of the composition of DES, the different DESs (ChCl-EG, ChCl-GI, ChCl-BDO) with the same volume ratio of DES:H2O (6:4) was investigated. Thus, the higher phenolic compounds extraction in Figure 2(a) and Figure 2(b) was the similar value. Through the preliminary experiments, Choline chloride (ChCl) as a kind of hydrogen bond acceptor, polyol as hydrogen bond donor showed a better extraction efficiency of phenolic acid compounds. Thus, in this study, the polyol (ethylene glycol (EG), glycerol (GI) and 1,4-butanediol (BDO)) as hydrogen bond donor was screened. And the extraction yield of the total phenol content as the response value, the different DESs (ChCl-EG, ChCl-GI, ChCl-BDO) with the same volume ratio of DES:H2O (6:4) was investigated. The results were showed in Figure 2. Thus, three different ChCl-based DESs were investigated for extraction of phenolic compounds. In Fig. 2a, the extraction yields when using ChCl-EG, ChCl-Gl, and ChCl-BDO with different molar ratios (1:1, 1:2, 1:3, 1:4, 1:5) are displayed. In all three DESs, hydrogen bonds are shared between the HBD (polyol) and chloride ions from ChCl (HBA), as showed in Fig.1S. , for polyols, there are more than two hydroxyl groups in the molecule that can form large molecules by forming intramolecular hydrogen bonds, and can also form intermolecular hydrogen bonds with ChCl. The more number of OH was provide by GI, when the same molar ratios of polyol., but the better water absorbent performance of GI may form the intermolecular hydrogen bonds, resulting in a decrease in the number of hydrogen bonds with ChCl. Compared with BDO, the steric effects to the formation of intermolecular hydrogen bonds with is ChCl smaller. Therefore, The the extraction yield of total phenolic compounds with ChCl-EG (6.97 ± 0.11 mg/g) was higher than that with ChCl-Gl (4.79 ± 0.13 mg/g) and ChCl-BDO (4.77 ± 0.14 mg/g), and 1:3 was the optimum molar ratio of ChCl and polyol.

Figure 3. Is there a significant difference in the solid-liquid ratios between solvents for extraction? Please add standard deviation to data in tables and figures.

Response: We checked the whole text, and the standard deviation to all data was added.

- Section 2.4.1. Are these extraction conditions (time and phenolic content) comparable to those presented in section 2.1? Why did authors not homogenize the extraction times used in all the extraction processes?

Response: In section 2.1, in order to analysis the chemical composition of rattan materials, “aqueous extracts” and “ethanolic extracts” obtained by traditional method were investigated. It takes longer extraction time to make sure the extraction is complete. In our study, homogenate-assisted vacuum-cavitation extraction method with DES was optimized. The optimum conditions are rattan raw materials 1:15 g/mL solid-liquid ratio mixed with DES solution, homogenate-assisted extraction 2.0 min, and then vacuum-cavitation extraction 25 min.

-Section 2.4.2. It is not clear why do authors use aqueous ethanol or methanol indistinctly for establishing different extraction conditions (times, water content, solid:liquid ratio, etc.) and then select them as the best extraction conditions because they resulted in higher extract contents, without considering the effect of solvent polarity and its effect of the phenolics solubility. There are several works in literature that have demonstrated that methanol shows significant differences when compared to ethanol or water solvents for phenolic compounds extraction. It is recommended to homogenize the solvents and matrices used for the establishment of the “best” or “optimum” extraction conditions.

Response: Thanks for reviewer’s constructive comment, 80% methanol as a compared solvent was used for homogenate extraction. But the data which was listed in Table 1S as Supplementary Material was not showed in this manuscript. The data listed in the text was all from “DES extracts” and “aqueous extracts”.

Table 1S Parameters of homogenate-assisted extraction

No.

Operation

solid-liquid ratio (g/mL)

duration time (min)

solvent

total phenolic content (mg/g)

1-1

Homogenate

1:15

2.0

DES

8.75 ± 0.16

1-2

Homogenate

1:15

2.0

Pure water

1.97 ± 0.04

1-3

Homogenate

1:15

2.0

80% methanol

6.44 ±   0.11

2-1

vacuum-cavitation

1:15

25.0

DES

90.33 ± 1.89

2-2

vacuum-cavitation

1:15

25.0

Pure water

21.44 ± 0.44

2-3

vacuum-cavitation

1:15

25.0

80% methanol

74.55 ± 1.23

3-1

ultrasonic-assisted

1:15

25.0

DES

84.32 ± 1.98

3-2

ultrasonic-assisted

1:15

25.0

Pure water

17.25 ± 0.64

3-3

ultrasonic-assisted

1:15

25.0

80% methanol

64.83 ± 1.14

-Figure 4. Discussion in section 2.4.2 only indicate the effect of aqueous methanol or water on the total phenolic content, but in Figure 4 appears two profiles corresponding to aqueous extracts and DES extracts. The discussion about DES extraction is not contained in this section. In agreement to section 2.1 where 6 h were used for extraction, why was a maximum of 3 min used for the rest of the extraction methods? Then, results in section 2.1 can be comparable to those shown in the rest of the document?

Response: In section 2.1, in order to analysis the chemical composition of rattan materials, “aqueous extracts” and “ethanolic extracts” obtained by traditional method were investigated. It takes longer extraction time to make sure the extraction is complete. In our study, homogenate-assisted vacuum-cavitation extraction method with DES was optimized. The optimum conditions are rattan raw materials 1:15 g/mL solid-liquid ratio mixed with DES solution, homogenate-assisted extraction 2.0 min, and then vacuum-cavitation extraction 25 min. The data listed in the text was all from “DES extracts” and “aqueous extracts”. Figure 3 has been replaced.

-Section 2.4.3. It is stated “During homogenization process, the thickness ….” What homogenization conditions were used in this section? They were not specified in M&M section.

Response: The optimum conditions obtained in subsections 2.4.1 and 2.4.2 apply to the experimental method of section 3.2.4 to homogenate-assisted extraction of phenolic compounds, which were the homogenate-assisted extraction conditions.

-Figure 5. Please add the transport phenomena signaling and improve the transport phenomena discussion, indicate if under the extraction conditions described if the diffusion mechanism or convective resistance to mass transfer results in the most important phenomena that control the phenolic compounds extraction.

Response: The transport phenomena discussion in section 2.4.3 has been improved.

-Section 2.5. Authors presents a lot of experiments, I consider that a good design of experiments could be very helpful for elucidating the most important parameters and variables that affect the phenolic content in extracts (solvent, time, cavitation, vacuum, etc).

Response: The optimum parameters were listed in Table 1S.

-Section 2.5.2. This section is merely descriptive about cavitation phenomena, it does not include results information, it could be incorporated in another section that already includes results data.

Response: Some discussion was added in section 2.5.2.

-Section 2.6. It should be convenient to indicate which extraction conditions were stated by authors as the “optimum extraction conditions”. Please do not repeat information within figures and main text. What is the relevance for this work when authors differentiate the phenolic contents in the diverse plant tissues? In agreement to introduction section, most of these tissues are discarded as byproducts, why do authors not only blend all the plant wastes and just focused this document for defining the use of DES and non-conventional extraction process for phenolics yield from rattan plant?

Response: To avoid repetition, Fig. 10 and section 2.8 were deleted. In order to make better use of abandoned resources, the phenolic compounds content in the diverse plant tissues were investigated. The phenolic compounds, well known as the thermosensitive substances, were applied in the field of food additives, colorants, and antioxidants. Therefore, the green extraction solvent and an extraction method at room temperature were key factors for phenolic compounds’ separation. In this study, a method of homogenate-assisted vacuum-cavitation extraction with DES was developed.

-Section 2.7. It is recommended to show the chromatograms obtained from different plant tissues, in order to show if all the tissues present the same trend of phenolics or there is a dependence of the type of phenolics with the tissues where they were extracted.

Response: The outer chromatogram in Figure 1 is the HPLC chromatogram of rattan blade sample, which has the most phenolic acid species (6 kinds of phenolic compounds). In other samples, there are less phenolic acid species (less peaks) and content (less peak area) than the blade sample.

-Section 2.8. There is a misunderstood between the title of the section and the figure that shows the results information, this means that title includes the analysis of aqueous and ethanolic extracts, but figure 10 only includes DES and aqueous extracts, and at the end it is not clear what extraction conditions were used at this point.

Response: To avoid misunderstood and repetition, Fig. 10 and section 2.8 were deleted.

-Section 3.1.1. Indicate if the whole rattan plant was purchase in the same place, time, supplier, etc. Indicate the year of the purchase.

Response: Yes, the whole rattan plant was collected in the same place in October 2016, and this information was added in section 3.1.1.

-Table 2. Information in this table is already described in the text.

Response: The duplicate information in the text has been deleted.

-Section 3.2.1. indicate if a gallic acid standard curve was used.

Response: Precision the standard 10.0 mg, and then set capacity to 10 mL, obtained the single standard reserve solution. And then mixed the single standard reserve solution, obtained the mixing standards reserve solution (0.167 mg/mL for each phenolic standard). Finally, diluted the mixing standards by 2 times in turn, and The six phenolic compounds were detected by measuring absorbance at 274 nm during a run time of 75 60 min (Fig. 1). Table 1 shows the retention time, the corresponding calibration curves, the limit of quantity (LOQ) and the limit of detection (LOD) for gallic acid, protocatechuic acid, protocatechuic aldehyde, (+)-catechins, chlorogenic acid, and EGCG. The retention time of gallic acid is 5.2 min, and the standard curve of gallic acid is Y=3.07×107X+4.01×104 (LOQ: 0.0157-0.1672 mg/mL), R2=0.9999. After detecting the peak area of 5.2 min in Rattan samples as Y value, calculated the X value (the concentration of gallic acid, mg/mL), and then multiplied the volume of the DES extracts (mL), divided by the mass of the raw material (g),obtained the extraction yield of gallic acid (mg/g).

- Section 3.2.3. indicate the mixing time for DES preparation and which was the criteria used for establishing the DES formation.

Response: DESs were synthesized by a heating method [36]. The hydrogen bond acceptor in each DES was ChCl. This was mixed with a hydrogen bond donor (one of the polyols listed in 2.3.1) in an appropriate molar ratio and placed in a reaction flask with magnetic agitation at 80 °C for 30 min, resulting in the formation of a transparent, homogeneous liquid. This information was added in section 3.2.3.

-Section 3.2.4, Temperature of extraction is not declared, does it affect the extraction yield? What was the particle size and polydispersity in plant material achieved under this homogenization process? Was it the same for all the plant tissues? Could particle size differences affect the phenolics extraction?

Response: The homogenate-assisted and vacuum-cavitation process was operated at room temperature, which was suitable for the thermosensitive substances extraction. In homogenate-assisted extraction process, shearing of solid particles into smaller pieces, the particle size of raw material is getting smaller and smaller with the homogenate time increasing. The shearing action reduces the size of the large solid particles and destroys most of the plant cell walls. At the same time, a liquid membrane layer which contained more hydrogen bonds forms at the cell surface, the permeability of cell was increased and the cell contents phenolic acids are released into the solvent. The raw materials for homogenate-assisted extraction are dried rattan blade (sheared into small pieces of 1 cm×1 cm square pieces). This explanation was added in section 3.1.1.

-In M&M section is not clear what procedures were done with ethanol, methanol or water like solvents in order to establish the optimum conditions for extraction. Also, how do authors ensure the finding of “optimum extraction conditions” if there is not any statistical analysis done?

Response: In section 2.1, in order to analysis the chemical composition of rattan materials, “aqueous extracts” and “ethanolic extracts” obtained by traditional method were investigated. It takes longer extraction time to make sure the extraction is complete. In our study, homogenate-assisted vacuum-cavitation extraction method with DES was optimized. The optimum conditions are rattan raw materials 1:15 g/mL solid-liquid ratio mixed with DES solution, homogenate-assisted extraction 2.0 min, and then vacuum-cavitation extraction 25 min. The data listed in the text was all from “DES extracts” and “aqueous extracts”. 80% methanol as a compared solvent was used for homogenate extraction. But the data which was listed in Table 1S as Supplementary Material was not showed in this manuscript.

-It is highly recommended to use specialized statistical software for a more rigorous analysis like means comparisons tests, surface response methodology, design of experiments.

Response: Thanks for reviewer’s constructive comment, in our study, there is two steps in the extraction process. The homogenate-assisted process followed by the vacuum-cavitation process, and there are two parameters (solid-liquid ratio and homogenate time) in the homogenate-assisted process and cavitation time in vacuum-cavitation process. The optimized software, such as surface response methodology was not used less than 3 parameters. And the selection of DES is not the factor in this process. Thus, the single factor experiment was designed in our work. The optimum parameters of homogenate-assisted vacuum-cavitation extraction with different solvent were list in Table 1S.

-Section 4. In agreement to chromatograms, there are at least 4 peaks signals with higher areas than the standard compounds used, it is recommended to make the identification of these compounds since they may affect in bigger proportion the effect on phenolics quantification.

Response: Yes, we have also noticed these peaks, but according to our existing standards (common phenolic acid), this peak was not identified. It may be not a phenolic acid compound. In my further experiment, I'm trying to identify the compound through mass spectrometry and nuclear magnetic resonance. And it will be published in another manuscript. Thanks for reviewer’s constructive comment, in this study, we mainly studied the extraction method with deep eutectic solvents and the content difference of phenolic acid compounds, such as gallic acid, protocatechuic acid, protocatechuic aldehyde, (+)-catechins, chlorogenic acid, and EGCG in different organs.

-Because formation of DES depends on equilibrium relationships, how does the water addition affect such equilibrium and therefore their functionality for improving the phenolics recovery?

Response: In our study, the addition of water decreased the viscosity of DES, and increased the osmotic effect of solvent, furthermore, enhanced the mass transfer from the plant matrices to solution. On the other hand, the addition of water increased the chemical polarity of DES, and the dissolve of phenolic compounds increased according to the “similar solubility principle”. This explanation was added in section 2.3.2.

-Indicate standard deviation values for quantities described along the complete document.

Response: Results are expressed as mean values ± SD (n = 3). Mean values from different experiments were compared by ANOVA using Microsoft Excel software. Differences at p < 0.05 were considered to be significant. The standard deviation values for quantities described along the complete document have been added.

It seems to be a very broad document with a lot of important information, its handle sometimes could result in a very complicated task, where all the activities should be interrelated among each other and provide trusty information. In this sense, it is recommended to authors to be clearer in the presentation, discussion and interrelationship of information, selecting the mains effects for achieving all the objectives raised.

Response: We combed through the whole document carefully, and readjusted the important information, make the discussion and interrelationship more clearly.

Reviewer 4 Report

-          Τhe paragraph below is very general and should be replaced with a paragraph with more specific information regarding polyphenols of the certain plant. This information should be connected with the selection of the polyphenols studied in the manuscript.

 Phenolic compounds can be divided into several classes, including phenolic acids and their derivatives, dehydrodiferulates and dehydrotriferulates, flavan-3-ol monomers, and others [24]. Most phenolic compounds contain multiple hydroxyl groups. As food additives and colorants, many studies have reported on the antioxidant [25-27] and free-radical scavenging activities [28-30] of phenolic compounds, which can be attributed to the phenolic hydroxyl groups. Thus, a rapid, simple, low-temperature method to extract phenolic compounds from rattan waste materials would provide considerable practical opportunities

-          In my opinion Figure 10 should be replaced with a table or graphs, I believe that pie graphs in not proper diagram for results of chemical analysis.

-          Conclusion should be more general without mentioning certain results

Author Response

Phenolic compounds can be divided into several classes, including phenolic acids and their derivatives, dehydrodiferulates and dehydrotriferulates, flavan-3-ol monomers, and others [24]. Most phenolic compounds contain multiple hydroxyl groups. As food additives and colorants, many studies have reported on the antioxidant [25-27] and free-radical scavenging activities [28-30] of phenolic compounds, which can be attributed to the phenolic hydroxyl groups. Thus, a rapid, simple, low-temperature method to extract phenolic compounds from rattan waste materials would provide considerable practical opportunities

Response: Thanks for reviewer’s constructive suggestion, the information associated with polyphenols studied in the manuscript has been added in Introduction.

-In my opinion Figure 10 should be replaced with a table or graphs, I believe that pie graphs in not proper diagram for results of chemical analysis.

Response: Figure 10 and section 2.8 have been deleted. But the contents of six phenolic compounds in aqueous extracts and in DES extracts from different parts of rattan plant were showed in Fig.9. And in Fig.10, from the point of view of a single phenolic acid, a distribution percentage was investigated. Although the data is repeated, the expression angle is different. It was clear described which organ has the highest content of a single phenolic compound (distribution percentage was marked by different colours). If Fig.10 (and section 2.8) was necessary, they will be added in the text again.

-Conclusion should be more general without mentioning certain results

Response: Conclusion has been refined.

Round 2

Reviewer 2 Report

The manuscript "Green Extraction of Six Phenolic Compounds from Rattan (Calamoideae faberii) with deep eutectic solvent by homogenate-assisted vacuum-cavitation method" was significantly improved compared to the previously submitted version. The authors responded to all the suggestions of the reviewer and accordingly added all changes in the text. In addition, Fig. 10 was removed and were added Fig. 1S, Fig. 2S, Tab. 1S and new Refs. 31 and 32.

I have an only small suggestion related to Fig. 9: y-axis title "Total phenolic content (mg/g)" should be replaced with "Phenolic compounds (mg/g)" in Fig. 9a and Fig 9b.

Conclusion:

This manuscript can be accepted after the implementation of suggested correction.